# A Genetic Evaluation System for New Zealand White Rabbit Germplasm Resources Based on SSR Markers

**DOI:** 10.3390/ani10081258

**Published:** 2020-07-24

**Authors:** Jiali Li, Bin Zhao, Yang Chen, Bohao Zhao, Naisu Yang, Shuaishuai Hu, Jinyu Shen, Xinsheng Wu

**Affiliations:** 1College of Animal Science and Technology, Yangzhou University, Yangzhou 225009, China; li1193117036@163.com (J.L.); xiaofengzhao0308@163.com (B.Z.); yangc@yzu.edu.cn (Y.C.); zhao598841633@163.com (B.Z.); yangnaisu@foxmail.com (N.Y.); 18852726848@163.com (S.H.); 13665202109@163.com (J.S.); 2Joint International Research Laboratory of Agriculture & Agri-Product Safety, Yangzhou University, Yangzhou 225009, China

**Keywords:** rabbit, microsatellite, genetic diversity, conservation effect

## Abstract

**Simple Summary:**

The New Zealand white rabbit (*Oryctolagus cuniculus*) is one of the most important breeds of commercial and experimental rabbits in the world, and also one of the most raised rabbit breeds in China. Our goal was to develop a suite of microsatellite markers to aid future conservation genetics research for the *Oryctolagus cuniculus* breeds. Based on the genetic diversity of 130 New Zealand white rabbits, we obtained a set combination of 22 markers. Then, we performed a genetic analysis of 200 New Zealand white rabbits corresponding to two generations with this combination. It can be used to evaluate the breed conservation of rabbit germplasm resources.

**Abstract:**

At present, there is an abundance of quality domestic rabbit breeds in China. However, due to the lack of technical standards for the genetic evaluation of rabbit germplasm resources, there have been a number of problems, such as poor breed conservation. By studying the genetic diversity of 130 New Zealand white rabbits (regardless of generation), we obtained the best simple sequence repeat (SSR) marker combination. We found that, when using microsatellite markers for the effective genetic evaluation of domestic rabbits, the number of records should be greater than 60 and the marker number more than 22. Through the comparative analysis of 30 combinations of 22 markers, the optimal combination of 22 markers was determined, and the 22 SSR polymorphic loci were distributed on different chromosomes. We performed a genetic analysis of 200 New Zealand white rabbits corresponding to two generations, using the best SSR polymorphic loci combination. There were no significant differences in the genetic diversity parameters between the two generations of rabbits (*p* > 0.05), indicating that the characteristics of this excellent rabbit germplasm have been effectively preserved. At the same time, we verified that the established method can be used to evaluate the breed conservation of rabbit germplasm resources.

## 1. Introduction

Currently, there is a wealth of domestic rabbit breeds in China, but, due to the lack of standards for the genetic evaluation of rabbit germplasm resources, there have been numerous problems, including the incomplete conservation of breeds. There is insufficient awareness of the necessity to protect the genetic resources of rabbit breeds that are not increasing in numbers, and the current resource protection measures are inadequate. Various factors have led to a sharp decrease in the number of local rabbit breed resource groups in China, and some local rabbit breeds are on the brink of extinction, e.g., Sichuan white rabbits, Himalayan rabbits, and Yunnan flower giant rabbits, and their genetic footprints are now difficult to trace.

In current molecular genetics research, the main DNA markers used are microsatellites (simple sequence repeats, SSRs) [1], restriction fragment length polymorphisms [2], amplified fragment length polymorphisms [3], expressed sequence tags, single nucleotide polymorphism markers [4], mitochondrial DNA [5], and other technologies. Among these, SSRs are ideal for investigating population genetics because of their high numbers of polymorphisms, wide genomic distributions, codominant inheritance, and high degree of reproducibility [6,7,8]. The application of microsatellite markers in animal breeding can significantly contribute to the overall health and disease resistance of farm animals [9]. Rabbit microsatellite research began at the end of the 20th century [10], and four sites—*SOL03*, *SOL28*, *SOL08*, and *SOL30*—were found using cloning techniques. The genetic analysis of these four loci in 20 wild rabbits from the East Angola region was used to the advantage of their hybridization. Van et al. [11] found 181 SSR markers in the genetic map of rabbits by searching the EMBL genomic database, and distributed microsatellite loci at 2–3 kb intervals on the genome.

Seven SSR polymorphic loci (*SOL03*, *SAT2*, *SOL30*, *SOL33*, *SAT5*, *SAT8*, and *SAT12*) were used to analyze the population structure of European hares from 20 regions [12], and the results showed that there had been a large-scale outbreak of myxomatosis in Europe. Ten SSR polymorphic loci (*SOL03*, *SOL08*, *SOL28*, *SAT5*, *SAT7*, *SOL30*, *SOL33*, *SOL44*, *SAT8*, and *SAT12*) were used to investigate 17 randomly selected hare species from the East Angola region for four consecutive years [13]. The hares showed genetic separation within and between populations and did not follow the Hardy–Weinberg equilibrium. Zenger et al. [14] conducted genetic diversity studies on seven SSR polymorphic loci of five Australian rabbits. The genetic bottleneck effect was not apparent in the 13 rabbits introduced to France; therefore, the European hare population had maintained good genetic diversity.

SSR markers were usually used by researchers for genetic diversity and population structure [15,16,17,18] but less for conservation [19]. In this study, we used New Zealand white rabbits, one of the most important breeds of commercial [20] and experimental [21,22] rabbits in the world, as subjects in a genetic evaluation of rabbit germplasm resources. Our goal was to develop a panel of microsatellite markers to aid future conservation genetics research [23] for the New Zealand white rabbit as well as other *Oryctolagus cuniculus* breeds. We aimed to provide a record of the genetic diversity, genetic background data and origin differentiation of rabbit germplasm resources in China. We designed the study to provide a basis for the protection and rational scientific utilization of rabbit germplasm resources in the country.

## 2. Materials and Methods

### 2.1. Animals and DNA Extraction

This research was carried out in accordance with the recommendations of the Animal Care and Use Committee at Yangzhou University. The experimental procedures was approved by the Animal Care and Use Committee at Yangzhou University (Yangzhou, China, 15 September 2017, No. 201709003). Operational procedures were stringently conducted in accordance with the Laboratory Animal Requirements of Environment and Housing Facilities (GB14925-2001). All rabbits were from the Pizhou Dongfang Breeding Rabbit Company. We obtained fresh ear tissue from 330 New Zealand white rabbits. DNA was extracted from fresh ear tissue using TIANamp Genomic DNA kits (Tiangen Biotech, Beijing, China). The DNA was analyzed using agarose gel electrophoresis, and the concentration and purity were checked with a NanoDrop 1000 (Thermo Scientific, Wilmington, CA, USA).

### 2.2. Microsatellite Genotyping

To genotype all the subjects, we used the double PCR approach. By referring to the rabbit gene database and existing references [10,11,24,25], 43 SSR marker synthesis primers were screened, and the 43 loci were distributed throughout 18 pairs of chromosomes. First, we used unlabeled SSR primers. PCRs were performed in 20 μL volumes: 2 μL of l0 × PCR (Mg^2+^ free) Buffer, 0.8–1.2 μL of 25 mmol/L MgCl_2_, l.0 μL of 2.5 mmol/L dNTPs, 1 μL of each 10 μmol/μL unlabeled upstream and downstream primer, 1.0 U of Taq DNA polymerase, and 1.0 μL of 100 ng/μL DNA template, supplemented with 20 μL of ddH_2_O. The PCR cycles started with 5 min of denaturation at 95 °C, which was followed by 34 cycles of 45 s of denaturation at 94 °C, 40 s of annealing at 50–64 °C (depending on the loci), and 40 s of elongation at 72 °C; the program was ended with a 10 min final elongation at 72 °C.

The PCR products were examined by 1% agarose gel electrophoresis with observation under a UV lamp and photography to record the results. Then, the PCR products were detected using polyacrylamide gel electrophoresis (PAGE), and the ImageQuant300 imaging system was used to photograph and store the most effective images. Using the ONED scan software and referring to the PBR322 DNA/Msp I Marker, the PAGE bands were analyzed to determine the approximate location range of the 43 SSR loci, which laid the foundations for subsequent sequencing analysis.

According to the genotypic analysis results from the New Zealand white rabbit PAGE and referring to the sequence sizes of the target fragments of the SSR loci published in GenBank, the SSR loci were used as the parameters, and markers with a difference of more than 20 bp were used. The markers were combined, and each group had three loci. The 5’ ends of each pair of primers were labeled with FAM (blue), HEX (green), or TAMRA (yellow) fluorescent dyes, respectively, resulting in 15 different combinations. Short tandem repeat typing was performed according to the mixed PCR product of each labeled combination. The combination of 43 pairs of fluorescently labeled SSR-labeled primers and PCR reaction conditions are shown in Appendix A.

PCRs were performed in 20 μL volumes consisting of 2 μL of l0 × PCR (Mg^2+^-free) buffer; 0.8–1.2 μL of 25 mmol/L MgCl_2_; l μL of 2.5 mmol/L dNTPs; 1 μL of each HEX-, FAM-, or TAMRA-labeled forward and unlabeled reverse primer (10 μmol/µL); 1.0 U of Taq DNA polymerase; 1 μL DNA template (100 ng/μL); and 20 μL of ddH_2_O. The entire loading procedure was carried out on ice, and amplifications were carried out in a 96-well plate. Because the fluorescent SSR primers decompose easily, they were protected from light as much as possible during the sample loading. After the PCR amplifications, each set of amplification products was mixed then covered with tinfoil to prevent exposure to light during storage at −20 °C. The PCR amplification procedure was the same as that for the first PCR.

After 1% agarose gel electrophoresis examination, each of the three pairs of well-mixed fluorescent PCR products was sent to Suzhou Thermo Scientific Technology Co., Ltd. for final STR typing. The total volume of the sample was 13 μL, containing 3 μL of mixed PCR products, 10 μL of Hi-Di Formamide, and 0.5 μL of the GS-500 size standard. The Hi-DiG and GS-500 size standard solutions were uniformly mixed, the mixed solution was added to the original 96-well plate, and the same group of fluorescent PCR amplification products were mixed in equal amounts: one by one according to the numbered order. The sample was added dropwise to a 96-well plate and vortexed and denatured, and the treated sample was electrophoresed and detected.

### 2.3. Data Analysis

After the electrophoresis of the mixed sample, the data were processed with Genemapper 4.0. Pure, and heterozygous genes could be identified by the peak patterns. The Hardy–Weinberg Equilibrium value was estimated using default Markov chain parameters [26,27]. We estimated the effective population size based on the genotype linkage disequilibrium principle.

The allele frequency, number of effective alleles (Ne), expected heterozygosity (He), observed heterozygosity (Ho), and polymorphic information content (PIC) were analyzed with GENEPOP 4.2 [28,29].

## 3. Results

### 3.1. Determination of SSR Fluorescent Labeling Diversity by Capillary Electrophoresis

All the alleles of the 43 SSR loci were detected within 500 bp, with a detection range of between 88 and 456 bp. The agarose electrophoresis results for the PCR amplification products are shown in Appendix A. Appendix A shows the results of the capillary gel electrophoresis of the three SSR loci, *SAT2* (HEX), *SOL62* (FAM), and *INRACCDDV0313* (TRAMER).

Using the results of the microsatellite fluorescent labeling capillary electrophoresis, the genetic diversity parameters of New Zealand white rabbit population were analyzed. The results show that a total of 184 alleles were detected in 130 individuals with 43 SSR polymorphic loci (Table 1). The average number of alleles (NA) and effective number of alleles (Ne) were 4.163 and 1.974, respectively. The minimum expected heterozygosity (He) was 0.008, the maximum value was 0.709, and the average value was 0.450; the observed heterozygosity distribution range was 0.008–0.985, and the average value was 0.506; the minimum polymorphic information content was 0.008, the maximum was 0.684, and the average was 0.385.

### 3.2. Hardy–Weinberg Equilibrium Test and Effective Population Size Calculation

The Hardy–Weinberg equilibrium test results for the New Zealand white rabbits are shown in Table 2. There were 31 SSR polymorphic loci that highly significantly (*p* < 0.05) deviated from Hardy–Weinberg equilibrium in 130 individuals. There were 24 SSR loci in the unbalanced state (*p* < 0.01), and 12 SSR loci (*p* > 0.05) followed the equilibrium state: *SAT8*, *D7UTR5*, *SAT2*, *INRACCDDV0007*, *INRACCDDV0018*, *INRACCDDV0313*, *INRACCDDV0160*, *D3UTR2*, *D6UTR4*, *12L1C2*, *SOL03*, and *5LIE8* (df = 1). The effective population size of New Zealand white rabbits, which was calculated using the linkage disequilibrium method, was found to be 56.4 (*CI*95% = 54.0–59.0) in 43 loci of 130 rabbits.

### 3.3. Statistical Analysis of New Zealand White Rabbit Genetic Diversity Using Different Numbers of Records

The Ne, He, and average PIC of New Zealand white rabbits was analyzed using different numbers of records (Figure 1A). The results show that the number of effective alleles in different marker number gradients showed a certain trend according to the number of records, i.e., when the number of records was 10–60, the average fluctuated, mostly between 1.766–1.971. When the number of records reached 60, the changes generally stabilized, and the average Ne was mostly between 1.780 and 1.960. The He in different label number gradients showed an association with the number of records, i.e., when the number of records was 10–60, the average He was observed to fluctuate, mostly between 0.379 and 0.460; when the sample size reached 60, the average He generally stabilized, and was mostly between 0.380 and 0.451. The PIC was associated with the number of records at different label number gradients, i.e., when the number of records was 10–60, the average PIC fluctuated greatly, between 0.312 and 0.387. When the number of records reached 60, the change stabilized, and was mostly between 0.310 and 0.382.

### 3.4. Statistical Analysis of New Zealand White Rabbit Genetic Diversity with Different Loci Numbers

By analyzing the effect of the loci numbers on the Ne, He, and PIC for each number of records of New Zealand white rabbits, the trend at each locus was revealed (Figure 1B). The results show that the average Ne increased as the loci numbers increased from 10 to 22, peaking at 22 loci. At 22 loci, the average Ne stabilized, and the average Ne remained at about 1.955. The average He followed a trend related to the loci number, i.e., the average He increased as the loci number increased from 10 to 22, and the average He started to stabilize at approximately 22 loci. He stabilized at about 0.450. Similarly, the average PIC increased as the loci number increased from 10 to 22, peaking at 22 and over. At the 22 loci number, the PIC began to stabilize, at around 0.381.

### 3.5. Statistical Analysis of Genetic Diversity of Different Combinations under the Optimal Loci Number

In order to reduce the cost of the experiment and improve the efficiency of the work, we first numbered the original 43 SSR markers in order from 1 to 43, as shown in Appendix A. Then, from the 43 SSR markers, 30 sets of 22 SSR polymorphic loci combinations were randomly selected from the Excel table. The average Ne, He, and PIC of the 30 loci combinations were evaluated (Appendix A). The aim was to screen out the 22 best representative SSR marker combinations to provide a basis for the protection and rational scientific utilization of rabbit germplasm resources. The specific results are shown in Table 3. By comparing different genetic diversity parameters, it was found that the average Ne and average PIC of Combination 2 were 1.978 and 0.379, respectively, which were close to the average value of the genetic diversity parameters of 43 loci. We preliminarily determined that Combination 2 could symbolically represent the genetic diversity results for the original 43 SSR markers.

The positions of the best combination were primarily located on different chromosomes. The positions of the best combination loci were: *L8B5* on Chromosome 7; *D7UTR5* and *SOL62* on Chromosome 6; *SAT2* and *INRACCDDV007* on Chromosome 5; *INRACCDDV0087* on Chromosome 8; *INRACCDDV0192* on Chromosome 2; *INRACCDDV0185* on Chromosome 16; *INRACCDDV0190* on Chromosome 18; *INRACCDDV0346* on Chromosome 13; *INRACCDDV0160* on Chromosome 9; *INRACCDDV0160* on Chromosome 9; *SOL44* on Chromosome 8; *12L4A1* on Chromosome 3; *6L1F10* on Chromosome 4; *6L3F8* on Chromosome 3; *D3UTR2*, *7L1B10*, and *19L1C5* on Chromosome 3; and *SOL33*, *SOL03*, and *12LIE11* on Chromosome 9.

### 3.6. Monitoring Genetic Diversity of New Zealand White Rabbits in Different Generations

Statistical analysis was performed on the genetic diversity parameters of 43 markers and 22 optimal markers for the original generation of New Zealand white rabbits (Table 2). The results show that the genetic diversity parameters detected by the 22 and 43 SSR markers were not significantly different (*p* > 0.05).

Statistical analysis was performed on the genetic diversity parameters of the 45 total and 22 optimal SSR polymorphic loci for the first generation of New Zealand white rabbits (Table 4). The NA, Ne, He, Ho, and PIC of the 43 and 22 SSR polymorphic loci of the F1 generation of New Zealand white rabbits were tested by the chi-squared analysis of mean differences. The results show that the genetic diversity parameters detected by the two sets of SSR polymorphic loci were not significantly different (*p* > 0.05).

### 3.7. Application of Rabbit Genetic Evaluation System

The genetic diversity of the two generations of New Zealand white rabbits was analyzed using the optimal combination of SSR polymorphic loci. Table 5 shows that the average NA, average Ne, average He, average Ho, and average PIC in the F0 generation were 3.318, 2.177, 0.483, 0.519, and 0.413, respectively. In the F1 generation, the average NA, average Ne, average He, average Ho, and average PIC were 3.409, 1.875, 0.416, 0.498, and 0.351, respectively. The analysis of the optimal SSR polymorphic loci for the two generations of New Zealand white rabbits showed that the genetic diversity parameters were not significantly different (*p* > 0.05); thus, the germplasm characteristics of the rabbit breeds were effectively preserved.

In order to further verify the reliability of the 22 optimal marker loci for evaluating the breeding effect of New Zealand white rabbits, we analyzed the parameters of the genetic diversity of the two generations of New Zealand white rabbits at the original 43 SSR polymorphic loci. As Appendix A shows, using the genetic evaluation system established in this study, there were no significant differences among the mean values of each genetic diversity parameter using the 22 optimal loci and the 43 original loci for the F0 and F1 generations of New Zealand white rabbits (*p* > 0.05). This provides evidence for the feasibility of using the 22 optimal loci combinations for the preliminary genetic evaluation of rabbits.

## 4. Discussion

The preservation of the maximum genetic diversity in a population is one of the main objectives within a breed conservation program [30]. When studying rabbit conservation, the focus should be on the analysis of the genetic diversity of different rabbit populations. There are currently three types of breed conservation methods: live-breed, frozen-breed, and biotechnology-based breed conservations. Currently, the main livestock conservation method used is live-breed conservation. Furthermore, gene bank live conservation not only provides a centralized backup and preserves excellent Chinese rabbit breeds but can also be used to conduct research on the use of the gene bank platform for local Chinese rabbit breed resources. The phenotype of microsatellite alleles with codominant inheritance can be better presented by genotype. The most frequent alleles in a population are the most primitive and conservative, and the lower frequency alleles are caused by mutations during the evolution of the population, so the polymorphic performance of microsatellites more objectively reflects the evolution of the population and conservation effect. The statistical analyses of Ne, He, and PIC show that the microsatellites used in this study were suitable for genetic diversity analysis [31].

This study aimed to establish an effective and fast genetic evaluation system for rabbits using different parameters for the number of records and loci number and to provide a reference for exploring the germplasm resources and genetic structure of rabbit breeds in China. The method used in this study for exploring the effect of different numbers of records and loci numbers on the population genetic structure has been applied to other animal populations. Ceccobelli et al. reduced the loci number from 16 to 12 to distinguish the local goat population, indicating that microsatellite markers are an inexpensive method for discriminating local livestock breeds [18]. A combination set of 12 markers can be used with high confidence for forensic purposes in Busha cattle [32]. The genetic diversity of sub-Saharan African goats was assessed using 19 microsatellite markers [29]. Parker et al. used 96 microsatellites of 414 dogs from 85 breeds to assign 99% of animals to their breed [33]. The genetic relationships between 61 dog breeds were investigated [34] using the sampling of 1514 animals and a panel of 21 microsatellite markers.

The genetic diversity of a germplasm population can be tested at regular intervals through rapid and efficient detection methods, and, after detection, the population’s germplasm resources can be protected through measures to maintain a rich genetic diversity. The Food and Agriculture Organization of the United Nations (FAO) used SSR polymorphic loci to study the genetic diversity of 14 species of livestock. The research led to the standard conditions for assessing quality germplasm resources: a marker number greater than 30 and a number of records greater than 50. Nei [35,36] pointed out that when considering the impact of genetic diversity indicators, we should set a different number of records according to the purpose of the study. Therefore, the choice of number of records should be highly valued.

When evaluating the genetic diversity of rabbit populations, proper experimental design is especially crucial. The choice of locus type and quantity is equally as important as the number of records. In this study, SSR polymorphic loci were used to study various trends in the genetic diversity parameters of New Zealand white rabbit breeds using different numbers of markers. The investigation showed that the Ne varied with the number of records and loci number, i.e., the Ae tended to be stable at 22–43 loci. Therefore, when the Ae of the New Zealand white rabbits was analyzed, more than 22 loci were selected. The degree of heterozygosity is expected to change with the number of markers for each number of records, i.e., the genetic diversity parameters should stabilize at 22–43 loci. Similarly, the PIC for each number of records showed the same trend associated with loci number. The PIC also tended to begin to stabilize at 22 or more loci, so when New Zealand white rabbit genetics were evaluated for PIC, the loci number selected was 22, but numbers above that are more appropriate. Barker [37,38] found that when insufficient marker loci were used to study the genetic diversity of the population, the results lacked reliability. Therefore, when objectively analyzing the excellent germplasm resources and genetic background of rabbits, it is particularly important to establish an effective loci number. We found that for the genetic evaluation of the New Zealand white rabbits, the optimal loci number is 22 or more.

## 5. Conclusions

In this study, a combination of 22 optimal SSR polymorphic loci were selected and used to detect genetic variation in different generations of New Zealand white rabbit for gene bank conservation. The results provide a theoretical reference for future rabbit conservation. We found that the difference between the genetic diversity parameters of two generations was not significant (*p* > 0.05). This shows that the gene bank’s preservation method has effectively preserved the germplasm characteristics of New Zealand white rabbit breeds.

For the effective genetic evaluation of rabbits using SSR markers, a record number greater than 60 and loci number of more than 22 are appropriate. The evaluation system based on an optimum combination of 22 loci provides a fast and effective method for the evaluation and establishment of excellent rabbit germplasms. The established method needs to be validated in an outside population of rabbits.

## Figures and Tables

**Figure 1 animals-10-01258-f001:**
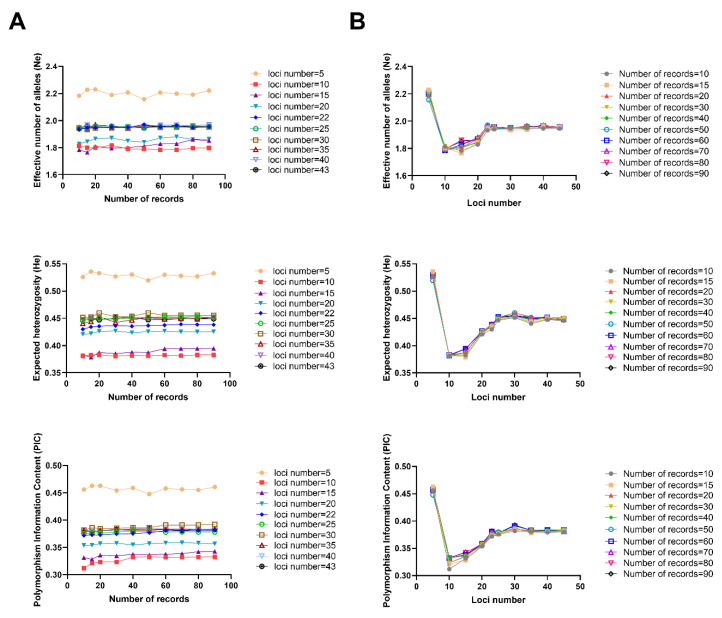
Statistical analysis of genetic diversity of New Zealand white rabbits under different numbers of records and loci numbers. (**A**) Changes in genetic diversity parameters with the number of records for different loci numbers for New Zealand white rabbits; (**B**) Changes in genetic diversity parameters along with loci numbers for different numbers of records for New Zealand white rabbits.

**Table 1 animals-10-01258-t001:** Number of alleles (NA), effective number of alleles (Ne), expected heterozygosity (He), observed heterozygosity (Ho), Polymorphism Information Content (PIC), and Hardy–Weinberg Equilibrium values (*p*-values) at the 43 best simple sequence repeat (SSR) loci.

Loci	NA	Ne	He	Ho	PIC	*p*-Value
*L8B5*	2.000	1.289	0.224	0.246	0.198	0.002
*SAT8*	4.000	1.928	0.481	0.531	0.404	n.s.
*D7UTR5*	8.000	2.914	0.657	0.669	0.596	n.s.
*SAT3*	2.000	1.589	0.371	0.385	0.301	0.004
*SAT4*	2.000	2.000	0.500	0.985	0.374	0.000
*SOL62*	2.000	1.762	0.432	0.415	0.338	0.000
*SAT2*	2.000	1.822	0.451	0.400	0.349	n.s.
*INRACCD* *DV0003*	3.000	1.142	0.125	0.146	0.118	0.005
*INRACCDDV0007*	3.000	1.142	0.125	0.146	0.118	n.s.
*INRACCDDV0010*	2.000	1.609	0.378	0.415	0.306	0.000
*INRACCDDV0087*	3.000	2.466	0.595	0.631	0.508	0.009
*INRACCDDV0018*	3.000	2.578	0.612	0.554	0.531	n.s.
*INRACCDDV0192*	2.000	1.991	0.498	0.442	0.373	0.000
*INRACCDDV0185*	3.000	1.359	0.264	0.254	0.231	0.036
*INRACCDDV0190*	2.000	1.518	0.341	0.323	0.282	0.000
*INRACCDDV0152*	2.000	1.599	0.375	0.400	0.304	0.004
*INRACCDDV0309*	3.000	1.858	0.462	0.315	0.412	0.001
*INRACCDDV0313*	4.000	2.743	0.635	0.515	0.562	n.s.
*INRACCDDV0314*	3.000	1.172	0.147	0.169	0.141	0.000
*INRACCDDV0346*	2.000	1.548	0.354	0.462	0.291	0.006
*INRACCDDV0160*	2.000	1.830	0.454	0.685	0.350	n.s.
*INRACCDDV0157*	6.000	1.606	0.377	0.385	0.328	0.014
*SOL44*	7.000	2.154	0.536	0.515	0.425	0.022
*12L4A1*	8.000	1.927	0.481	0.415	0.456	0.048
*6L1F10*	10.000	3.431	0.709	0.600	0.684	0.001
*6L7C11*	10.000	2.615	0.618	0.538	0.557	0.042
*6L3F8*	9.000	2.882	0.653	0.415	0.599	0.005
*6L2H3*	6.000	1.459	0.314	0.269	0.281	0.000
*D3U* *TR2*	6.000	1.251	0.201	0.169	0.191	n.s.
*7L1B10*	5.000	2.203	0.546	0.462	0.488	0.000
*D6UTR4*	9.000	2.310	0.567	0.485	0.535	n.s.
*SOL08*	3.000	1.797	0.442	0.400	0.346	0.025
*12L1C2*	4.000	1.939	0.484	0.500	0.380	n.s.
*SAT12*	3.000	1.943	0.485	0.508	0.377	0.001
*19L1C5*	2.000	1.008	0.008	0.008	0.008	0.000
*SOL33*	3.000	2.618	0.618	0.538	0.537	0.000
*12L5A6*	3.000	1.973	0.493	0.462	0.374	0.028
*SAT7*	3.000	1.919	0.478	0.454	0.370	0.000
*SOL03*	3.000	2.136	0.532	0.515	0.447	n.s.
*SOL30*	5.000	2.349	0.574	0.485	0.478	0.005
*5LIE8*	7.000	2.005	0.501	0.400	0.396	n.s.
*12LIE11*	4.000	2.695	0.629	0.546	0.557	0.000
*SAT5*	4.000	2.652	0.623	0.638	0.553	0.003

n.s.: not significant.

**Table 2 animals-10-01258-t002:** Genetic diversity analysis of F0 generation of New Zealand white rabbits with two SSR marker combinations.

Genetic Diversity Parameter	43 Loci	22 Loci	*p*-Value
NA	3.370 ± 1.915	3.318 ± 1.729	0.974
Ne	2.100 ± 0.785	2.177 ± 0.833	0.913
He	0.473 ± 0.151	0.483 ± 0.171	0.943
Ho	0.497 ± 0.202	0.519 ± 0.231	0.907
PIC	0.401 ± 0.144	0.413 ± 0.163	0.928

**Table 3 animals-10-01258-t003:** Genetic diversity of 30 combinations of 22 SSR markers.

Group ^1^	Ne	He	PIC
C1	2.031	0.481	0.406
C2	1.978	0.439	0.379
C3	1.976	0.451	0.382
C4	2.090	0.493	0.418
C5	1.933	0.440	0.377
C6	1.892	0.440	0.370
C7	2.000	0.462	0.400
C8	2.024	0.467	0.396
C9	2.016	0.463	0.394
C10	1.906	0.435	0.367
C11	2.017	0.462	0.394
C12	1.885	0.412	0.352
C13	1.836	0.411	0.346
C14	2.023	0.476	0.400
C15	1.937	0.435	0.375
C16	1.992	0.436	0.380
C17	2.065	0.473	0.409
C18	1.982	0.462	0.386
C19	1.959	0.466	0.390
C20	1.913	0.442	0.370
C21	1.930	0.431	0.366
C22	1.928	0.430	0.369
C23	1.986	0.451	0.386
C24	2.022	0.481	0.406
C25	2.022	0.481	0.406
C26	1.904	0.444	0.370
C27	1.907	0.446	0.371
C28	2.084	0.478	0.412
C29	2.123	0.500	0.429
C30	1.940	0.441	0.374
43 SSR markers	1.974	0.450	0.385

^1^ C: combination.

**Table 4 animals-10-01258-t004:** Genetic diversity analysis of F1 generation of New Zealand white rabbits with two SSR marker combinations.

Genetic Diversity Parameter	43 Loci	22 Loci	*p*-Value
NA	4.047 ± 2.382	3.409 ± 1.817	0.731
Ne	1.969 ± 0.546	1.875 ± 0.573	0.847
He	0.449 ± 0.162	0.416 ± 0.182	0.826
Ho	0.506 ± 0.220	0.498 ± 0.265	0.970
PIC	0.384 ± 0.146	0.351 ± 0.159	0.804

**Table 5 animals-10-01258-t005:** Genetic diversity analysis of two generations of New Zealand white rabbit with the 22 optimal SSR markers.

Genetic Diversity Parameter	F0	F1	*p*-Value
NA	3.318 ± 1.729	3.409 ± 1.817	0.953
Ne	2.177 ± 0.833	1.875 ± 0.573	0.632
He	0.483 ± 0.171	0.416 ± 0.182	0.666
Ho	0.519 ± 0.231	0.498 ± 0.265	0.923
PIC	0.413 ± 0.163	0.351 ± 0.159	0.662

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
