# Peer review of "A Genetic Evaluation System for New Zealand White Rabbit Germplasm Resources Based on SSR Markers"

_animals, 2020, doi:10.3390/ani10081258_

Round 1

Reviewer 1 Report

No comments

Author Response

Response to Reviewer 1 Comments

Point 1: No comments.

Response 1: We appreciate for Reviewer’ warm work earnestly. Thank you very much for your comments and suggestions.

Reviewer 2 Report

Overall, the authors addressed several of my concerns.  However, when discussing the validity of the research outside of the Pizhou Dongfang Breeding Rabbit Company, the authors stated that their follow-up research from the Yuyaoxinnong Rabbit Farm confirmed their results.  As this material in not presented in the manuscript (i.e. will be published in a separate manuscript), then the authors need to add a sentence in the conclusion that the current research need to be validated in an outside population of rabbits.  In the current PDF format, figure 1 is still very difficult to read but this can be corrected in copy editing.

Author Response

Response to Reviewer 2 Comments

Point 1: Overall, the authors addressed several of my concerns.  However, when discussing the validity of the research outside of the Pizhou Dongfang Breeding Rabbit Company, the authors stated that their follow-up research from the Yuyaoxinnong Rabbit Farm confirmed their results.  As this material in not presented in the manuscript (i.e. will be published in a separate manuscript), then the authors need to add a sentence in the conclusion that the current research need to be validated in an outside population of rabbits.  In the current PDF format, figure 1 is still very difficult to read but this can be corrected in copy editing.

Response 1: We have made correction according to the Reviewer’s comments. The original figure 1 has been uploaded to the system and can be enlarged to see more clearly.

We appreciate for Reviewer’ warm work earnestly. Thank you very much for your comments and suggestions.

Reviewer 3 Report

The authors addressed most of the reviewers' concerns and made appropriate changes to manuscript. I think the questions raised were now satisfactorily answered.

Author Response

Response to Reviewer 3 Comments

Point 1: The authors addressed most of the reviewers' concerns and made appropriate changes to manuscript. I think the questions raised were now satisfactorily answered.

Response 1: We appreciate for Reviewer’ warm work earnestly. Thank you very much for your comments and suggestions.

This manuscript is a resubmission of an earlier submission. The following is a list of the peer review reports and author responses from that submission.

Round 1

Reviewer 1 Report

ABSTRACT
Abbreviation SSR should always be explained.
Sample size should also be changed to "number of records" or "informative number of individual". Otherwise it is rather confusing to the reader.

INTRODUCTION
LINE 40 - should be "there have been numerous problems"

LINE 47-71 - very detailed for introduction. I recommend to shorten this part and move more details on previous research to the discussion.

MATERIAL AND METHODS
Too much information is given in this section e.g. we do not need to know that Excel sheet was used. Please, read this section very carefully and keep only the information that is essential to understand the process of data preparation and its analysis.

I would recommend using terms "parental generation" or "F0" and "F1" to describe the two generations used in this work.

Provide sources of the equations used for calculations: Ae, He and PIC. How Allele frequencies were calculated?

Table 1 and Table 2 could be moved to supplementary material.

RESULTS
This section should be simplified by making the paragraphs more "to the point" instead of building on the volume of explanations as in discussion.

LINE 209-230 and 232-238 - are paragraphs for Methods and not results. Again too much details is given on the process of analysis e.g. Excel. The statistics used to interpret the results should be properly described in Methods section of the paper, including equations if needed and literature supporting use of such methods for this kind of data.

Figure 1 - very bad quality. The same parameter should be presented on the figures with the same scale to make it easier to compare the results. Also it is not easily described and it takes some time to understand what those figures represent.

Figure S3 - requires a significant improvement of the axis description. Now it is impossible to understand what it pretenses.

Table 4 - it is enough to say "Group" instead of "the group of No."

Table 5 and Table 6 and Table 7 - too much number after a decimal. P-value should be simply "n.s." as all of those values are not significant.

DISCUSSION
There is a lot of information and literature discussion given for the topic of the paper, but I am getting confused with it. It is hard to find where the results of this work are discussed and how conclusions are drawn.

LINE 319-321 - you cannot say that values are different, but the difference is not significant. Such result means it is not different.

CONCLUSIONS
It is not clear to me how the authors come to those conclusions.

Table S1 - It should be simply "A list of selected loci".

Table S2 and S3 - I do not understand what is presented in those tables. What is "C" in the name of the columns?

Author Response

Response to Reviewer 1 Comments

Point 1: ABSTRACT

Abbreviation SSR should always be explained.

Sample size should also be changed to "number of records" or "informative number of individual". Otherwise it is rather confusing to the reader.

Response 1: We are very sorry for our negligence of abbreviation SSR. We have explained it in abstract. As Reviewer suggested that “sample size” should be changed, we have changed it in article.

Point 2: INTRODUCTION

LINE 40 - should be "there have been numerous problems"

Response 2: We have made correction according to the Reviewer’s comments.

Point 3: INTRODUCTION

LINE 47-71 - very detailed for introduction. I recommend to shorten this part and move more details on previous research to the discussion.

Response 3: We have made correction according to the Reviewer’s comments.

Point 4: MATERIAL AND METHODS

Too much information is given in this section e.g. we do not need to know that Excel sheet was used. Please, read this section very carefully and keep only the information that is essential to understand the process of data preparation and its analysis.

Response 4: We have made correction according to the Reviewer’s comments.

Point 5: I would recommend using terms "parental generation" or "F0" and "F1" to describe the two generations used in this work.

Response 5: Considering the Reviewer’s suggestion, we have changed "F0" and "F1" to describe the two generations.

Point 6: Provide sources of the equations used for calculations: Ae, He and PIC. How Allele frequencies were calculated?

Response 6: Another reviewer suggested to remove the mathematical formulas for “Number of effective alleles”, Expected heterozygosity (He) and PIC. The reason is that the mathematical formulas are well known. And we included in the text the software used for analysis and the reference of the software used. The Allele frequencies were calculated by GENEPOP 4.2.

Point 7: Table 1 and Table 2 could be moved to supplementary material.

Response 7: We have made correction according to the Reviewer’s comments.

Point 8: RESULTS

This section should be simplified by making the paragraphs more "to the point" instead of building on the volume of explanations as in discussion.

Response 8: We have made correction according to the Reviewer’s comments.

Point 9: LINE 209-230 and 232-238 - are paragraphs for Methods and not results. Again too much details is given on the process of analysis e.g. Excel. The statistics used to interpret the results should be properly described in Methods section of the paper, including equations if needed and literature supporting use of such methods for this kind of data.

Response 9: We have deleted some sentences for methods according to the Reviewer’s comments.

Point 10: Figure 1 - very bad quality. The same parameter should be presented on the figures with the same scale to make it easier to compare the results. Also it is not easily described and it takes some time to understand what those figures represent.

Response 10: We reduce the classes and used 10 classes for the loci (5, 10, 15, 20, 22,25, 30, 35, 40, 43) to re-made the Figure 1.

Point 11: Figure S3 - requires a significant improvement of the axis description. Now it is impossible to understand what it pretenses.

Response 11: Considering the Reviewer’s suggestion, we have made correction.

Point 12: Table 4 - it is enough to say "Group" instead of "the group of No."

Response 12: We have made correction according to the Reviewer’s comments.

Point 13: Table 5 and Table 6 and Table 7 - too much number after a decimal. P-value should be simply "n.s." as all of those values are not significant.

Response 13: We reduced results to three decimal places in all tables and in the text. As Reviewer suggested that "n.s.", we have made correction.

Point 14: DISCUSSION

There is a lot of information and literature discussion given for the topic of the paper, but I am getting confused with it. It is hard to find where the results of this work are discussed and how conclusions are drawn.

Response 14: We have made correction according to the Reviewer’s comments.

In this article, the best SSR was first selected, and then used to detect the conservation effect of New Zealand white rabbits. So in the first part of the discussion, we discussed the methods and effects of conservation, and why we chose microsatellite for breed conservation. Then we discussed the number of records and loci number were important for establishing an effective and fast genetic evaluation system.

Point 15: LINE 319-321 - you cannot say that values are different, but the difference is not significant. Such result means it is not different.

Response 15: We have made correction according to the Reviewer’s comments.

Point 16: CONCLUSIONS

It is not clear to me how the authors come to those conclusions.

Response 16: We have re-written this part according to the Reviewer’s suggestion.

Point 17: Table S1 - It should be simply "A list of selected loci".

Response 17: We have made correction according to the Reviewer’s comments.

Point 18: Table S2 and S3 - I do not understand what is presented in those tables. What is "C" in the name of the columns?

Response 18: We are very sorry for our negligence of the abbreviation. “C” means “combination”. We have annotated below the table.

Special thanks to you for your good comments. These comments are all valuable and very helpful for revising and improving our paper, as well as the important guiding significance to our researches.

Reviewer 2 Report

Li et al. reported effective combination of the SSR markers for conservation of NZW rabbit germplasm resources. The outcome should be valuable for some specific readers, but the method used in this manuscript should be insufficient.

Author Response

Response to Reviewer 2 Comments

Point 1: Suggestions for Authors

Li et al. reported effective combination of the SSR markers for conservation of NZW rabbit germplasm resources. The outcome should be valuable for some specific readers, but the method used in this manuscript should be insufficient.

Response 1: We know that our method is inadequate, we will improve it in the following experiments. Our goal was to develop a panel of microsatellite markers to aid future conservation genetics research for the New Zealand white rabbit as well as other Oryctolagus cuniculus breeds. In our follow-up research, the best SSR marker combination was also used to evaluate the breed conservation effect of the Chinchilla rex rabbits in another company (Yuyaoxinnong Rabbit Farm). The genetic diversity parameters of Chinchilla rabbits were not significantly different among the F1, F4, F5 generations (P>0.05), indicating that the Yuyaoxinnong Rabbit Farm effectively preserved the genetic characteristics of the Chinchilla Rex rabbits. After that, we will use these SSR for verification with Angora rabbits in Anhui Province.

Special thanks to you for your good comments. These comments are all valuable and very helpful for revising and improving our paper, as well as the important guiding significance to our researches.

Reviewer 3 Report

The overall objective of this paper was to provide a concise set of microsatellite markers to assess breed conservation in New Zealand white rabbits.  Overall the paper is well laid out.  I just have a few issues:

The justification for using microsatellite for investigation of population genetics is based on publications from the late 80’s/early 90s.  Given the drastic advances in genetic technology, the authors need to discuss the limits of microsatellite markers and why they opted to choose this as their methodology.   

How are the authors correcting for multiple comparisons with statistical analyses?

Unfortunately the poor quality of figure1 made it very impossible to assess the data being presented.

Given that all rabbits were from the same breeding company, how does relatedness effect your results and how would this translate to New Zealand white rabbits outside of this company?

Author Response

Response to Reviewer 3 Comments

Point 1: The justification for using microsatellite for investigation of population genetics is based on publications from the late 80’s/early 90s.  Given the drastic advances in genetic technology, the authors need to discuss the limits of microsatellite markers and why they opted to choose this as their methodology.  

Response 1:

The application of RFLP technology is affected by genomes, probes and restriction enzymes.

SNPs are difficult to obtain in unknown species with poor molecular basic research and complete genome-wide sequencing. Therefore, it is mainly used in the study of humans and model animals.

SSR primers are difficult and expensive to obtain. SSR primers require molecular hybridization, screening positive clones, constructing sequence libraries, and detecting polymorphisms. They also require a lot of energy and expense.

SSR is widely distributed in the genome of eukaryotes, and the density of SSR distribution is large and uniform. SSR is highly polymorphic, conservative and versatile among species. Compared with other molecular markers, the markers are codominantly inherited, which has the advantages of easy identification, good detection repeatability, wide distribution in the genome, the ability to identify homozygotes and heterozygotes, etc.

We have added a small part to the discussion according to the Reviewer’s comments.

Point 2: How are the authors correcting for multiple comparisons with statistical analyses?

Response 2: The number of records and loci number are based on previous reports and experimental results. An article wrote“There are two sources of error in this estimation method, one is the error caused by the locus selection, and the other is the error caused by population sampling. Practical research shows that the sampling error of loci is much larger than that of population sampling. Therefore, when using microsatellite for population research, enough gene loci (minimum 20) and large enough population samples (at least 20) should be selected.”And another article wrote “ When the number of records is more than 50, the loci number is more than 25. When the number of records is more than 60, the loci number is more than 22.” However, Ceccobelli et al. reduced the loci number from 16 to 12 to distinguish the local goat population. Therefore, we could say the optimal loci number is 22 or more according to the trend of the Figure 1. But we couldn’t say the genetic analysis at the loci number = 16 can’t be used because they ( the loci number =16 and the loci number =22) are significantly different (P<0.05). We didn’t do multiple comparisons. If we need to make a statistical analysis, it should be multivariate analysis of variance.

The statistical analyses of 22 SSR and 43SSR is Paired-Samples T Test.

Point 3: Unfortunately the poor quality of figure1 made it very impossible to assess the data being presented.

Response 3: We used 10 classes for the loci: 5, 10, 15, 20, 22,25, 30, 35, 40, 43. We re-made the Figure 1.

Point 4: Given that all rabbits were from the same breeding company, how does relatedness effect your results and how would this translate to New Zealand white rabbits outside of this company?

Response 4: In our follow-up research, the best SSR marker combination was also used to evaluate the breed conservation effect of the Chinchilla rex rabbits in another company (Yuyaoxinnong Rabbit Farm). The genetic diversity parameters of Chinchilla rabbits were not significantly different among the F1,F4,F5 generations (P > 0.05), indicating that the Yuyaoxinnong Rabbit Farm effectively preserved the genetic characteristics of the Chinchilla Rex rabbits. At the same time, it also verified that the established genetic evaluation method can be used to evaluate the conservation effect of rabbit germplasm resources.

Special thanks to you for your good comments. These comments are all valuable and very helpful for revising and improving our paper, as well as the important guiding significance to our researches.

Reviewer 4 Report

The Authors analyzed the genetic variability on the New Zealand White rabbits bred in China. They aimed to develop a panel of microsatellite markers useful for future conservation plans for both New Zealand white rabbit as well as other Oryctolagus cuniculus breeds.

General comments

The topic of the paper is interesting for the readers, but the manuscript as such, at its present state, have to be improved.

Specific comments

Line 48: please remove “DNA”.

Line 50: please use only the acronym SSR

Line 57 to 71: an English revision of these sentences is needed.

Line 75: please replace “suite” with “panel”

Line 88: were the 330 animals of both sexes? What is the sex-ratio in the analyzed population?

Line 116: In the list of SSR, two are located on Chr X (INRACCDDV0256 and INRACCDDV0334). Why the Authors chosen this SSRs? How did they calculate the allele frequencies for these SSRs?

Line 136 to 158: the mathematical formulas for “Number of effective alleles”, Expected heterozygosity (He) and PIC are well known, so please remove them. Please, include in the text only the genetic diversity parameters estimated and the software used for analysis (i.e. GENEPOP or Microsatellite Toolkit or other). The reference of the software used have to be added to “Reference” section.

Line 141: please replace “Hardy-Weinberg Equilibrium was quantified” with “Hardy-Weinberg Equilibrium value was estimated”.

Please reduce results to three decimal places in all tables and in the text.

Please replace “A” with “NA” (Number of Alleles) and “Ae” with “Ne” (Effective number of alleles), “P” with “P-value” in all tables and in the text.

The value of observed heterozygosity (Ho) is reported in tables 5, 6 and 7, but not in table 3. Please add the values of Ho to the table3 and comment.

Table 3: please, modify the legend as “Table 3. Number of alleles (NA), effective number of alleles (Ne), expected heterozygosity (He), observed heterozygosity (Ho), Polymorphism Information Content (PIC) and Hardy-Weinberg Equilibrium values (P-value) at 45 SSR loci”.

Table 3 show the results of 39 SSR instead of 45 SSR. Please verify. Modify in the text and in the table: D7UTR5 with D7Utr5. Modify in the text: D3UTR2 with D3Utr2, D6UTR4 with D6Utr4.

Line 175 to 182: the Authors show the statistical results of 45 SSR, but the count for P-values is 33 SSR<0.05, 26 SSR<0.01 and 12 SSR>0.5, please verify.

Line 185-194: please clarify the sentences. In my opinion, the differences among classes are quite small and the variations are minimal. The great difference for the "5 loci" class is due to the presence of D7Utr5 (8 alleles) that falsifies the result.

Figure 1A: the graphical representation of trends of genetic diversity parameters correlated by sample size and number of loci have to be modified. This version of the figures is confused. In my opinion, is better to reduce the classes and, use only 9 classes for the loci: 5, 10, 15, 20, 25, 30, 35, 40, 45. Moreover, the quality of the figure have to be improved.

Line 209 to 221: the Authors have chosen the combination 2 as the most similar to original 45 SSR markers. In my opinion, the criterion for choosing the best combination of SSRs, for conservation purpose, has to be the polymorphism of loci and a high heterozygosity level. Why was not considered the combination C29 that shows the highest Ae, He and PIC?

Line 238: Figure 2A is missing.

Tables 4, 5, 6 and 7: please reduce line dimension.

Line 243: How many animals are the first generation made of? Are they of both sexes? What is the sex-ratio of the analyzed population?

Line 277: please replace “the local population” with “the local goat population”.

Line 317 to 322: please move to “Conclusion” section.

Line 330 to 336: please improve the legend of the supplementary tables and figures. The legend of the table have to be repeated in the first line of the corresponding table on the file excel.

Author Response

Response to Reviewer 4 Comments

Point 1: Line 48: please remove “DNA”.

Response 1: We have made correction according to the Reviewer’s comments.

Point 2: Line 50: please use only the acronym SSR

Response 2: We are very sorry for our incorrect writing. We have removed “microsatellite”.

Point 3: Line 57 to 71: an English revision of these sentences is needed.

Response 3: Considering the Reviewer’s suggestion, we have polished the entire article to the company.

Point 4: Line 75: please replace “suite” with “panel”

Response 4: We have made correction according to the Reviewer’s comments.

Point 5: Line 88: were the 330 animals of both sexes? What is the sex-ratio in the analyzed population?

Response 5: Yes, there were 165 male and 165 female, male : female =1:1.

Point 6: Line 116: In the list of SSR, two are located on Chr X (INRACCDDV0256 and INRACCDDV0334). Why the Authors chosen this SSRs? How did they calculate the allele frequencies for these SSRs?

Response 6: When we chose SSR, we wanted to cover all chromosomes as much as possible. We are very sorry for our negligence of Chr X. We decide to remove INRACCDDV0256 and INRACCDDV0334. All data are recalculated and analayzed again.

Point 7: Line 136 to 158: the mathematical formulas for “Number of effective alleles”, Expected heterozygosity (He) and PIC are well known, so please remove them. Please, include in the text only the genetic diversity parameters estimated and the software used for analysis (i.e. GENEPOP or Microsatellite Toolkit or other). The reference of the software used have to be added to “Reference” section.

Response 7: As Reviewer suggested that, we have removed the mathematical formulas and added reference.

Point 8: Line 141: please replace “Hardy-Weinberg Equilibrium was quantified” with “Hardy-Weinberg Equilibrium value was estimated”.

Response 8: We have made correction according to the Reviewer’s comments.

Point 9: Please reduce results to three decimal places in all tables and in the text.

Response 9: We have made correction according to the Reviewer’s comments.

Point 10: Please replace “A” with “NA” (Number of Alleles) and “Ae” with “Ne” (Effective number of alleles), “P” with “P-value” in all tables and in the text.

Response 10: We have made correction according to the Reviewer’s comments.

Point 11: The value of observed heterozygosity (Ho) is reported in tables 5, 6 and 7, but not in table 3. Please add the values of Ho to the table3 and comment.

Response 11: We have added the values of Ho to the table3 and comment according to the Reviewer’s comments.

Point 12: Table 3: please, modify the legend as “Table 3. Number of alleles (NA), effective number of alleles (Ne), expected heterozygosity (He), observed heterozygosity (Ho), Polymorphism Information Content (PIC) and Hardy-Weinberg Equilibrium values (P-value) at 45 SSR loci”.

Response 12: We have made correction according to the Reviewer’s comments.

Point 13: Table 3 show the results of 39 SSR instead of 45 SSR. Please verify. Modify in the text and in the table: D7UTR5 with D7Utr5. Modify in the text: D3UTR2 with D3Utr2, D6UTR4 with D6Utr4.

Response 13: We are very sorry for our negligence of 45 SSR. We have made correction according to the Reviewer’s comments.

Point 14: Line 175 to 182: the Authors show the statistical results of 45 SSR, but the count for P-values is 33 SSR<0.05, 26 SSR<0.01 and 12 SSR>0.5, please verify.

Response 14: We are very sorry for our negligence of the description, the P-values 33 SSR<0.05 are contained 26 SSR<0.01, 33 SSR(<0.05) +12 SSR(>0.5)=45 SSR.

Point 15: Line 185-194: please clarify the sentences. In my opinion, the differences among classes are quite small and the variations are minimal. The great difference for the "5 loci" class is due to the presence of D7Utr5 (8 alleles) that falsifies the result.

Response 15: It is really true as Reviewer suggested. There are two sources of error in this estimation method, one is the error caused by the locus selection, and the other is the error caused by population sampling. Practical research shows that the sampling error of loci is much larger than that of population sampling. Therefore, when using microsatellite to conduct population research, select enough loci (minimum 20) and large enough population samples (at least 20).

Point 16: Figure 1A: the graphical representation of trends of genetic diversity parameters correlated by sample size and number of loci have to be modified. This version of the figures is confused. In my opinion, is better to reduce the classes and, use only 9 classes for the loci: 5, 10, 15, 20, 25, 30, 35, 40, 45. Moreover, the quality of the figure have to be improved.

Response 16: We used 10 classes for the loci: 5, 10, 15, 20, 22,25, 30, 35, 40, 43. We added “22” because of we used 22 SSR markers in the article. We re-made the Figure 1.

Point 17: Line 209 to 221: the Authors have chosen the combination 2 as the most similar to original 45 SSR markers. In my opinion, the criterion for choosing the best combination of SSRs, for conservation purpose, has to be the polymorphism of loci and a high heterozygosity level. Why was not considered the combination C29 that shows the highest Ae, He and PIC?

Response 17: It is really true as Reviewer suggested that the combination C29 that shows the highest Ne, He and PIC. In our follow-up research, we will use C29 to evaluate the breed conservation effect of the Chinchilla rex rabbits in Yuyaoxinnong Rabbit Farm. Special thanks to you for your good comments.

Point 18: Line 238: Figure 2A is missing.

Response 18: We are very sorry for our negligence of Figure 2A, we have removed it.

Point 19: Tables 4, 5, 6 and 7: please reduce line dimension.

Response 19: As Reviewer suggested that, we have reduce line dimension.

Point 20: Line 243: How many animals are the first generation made of? Are they of both sexes? What is the sex-ratio of the analyzed population?

Response 20: The first generation were 65 male and 65 female. The sex-ratio of the analyzed population is 1:1.

Point 21: Line 277: please replace “the local population” with “the local goat population”.

Response 21: We have made correction according to the Reviewer’s comments.

Point 22: Line 317 to 322: please move to “Conclusion” section.

Response 22: We have made correction according to the Reviewer’s comments.

Point 23: Line 330 to 336: please improve the legend of the supplementary tables and figures. The legend of the table have to be repeated in the first line of the corresponding table on the file excel.

Response 23: We have made correction according to the Reviewer’s comments.

Special thanks to you for your good comments. These comments are all valuable and very helpful for revising and improving our paper, as well as the important guiding significance to our researches.
